# Preclinical Identification of Sulfasalazine’s Therapeutic Potential for Suppressing Colorectal Cancer Stemness and Metastasis through Targeting KRAS/MMP7/CD44 Signaling

**DOI:** 10.3390/biomedicines10020377

**Published:** 2022-02-04

**Authors:** Wai-Hung Leung, Jing-Wen Shih, Jian-Syun Chen, Ntlotlang Mokgautsi, Po-Li Wei, Yan-Jiun Huang

**Affiliations:** 1Division of Colon and Rectal Surgery, Department of Surgery, Mackay Memorial Hospital, No. 92, Sec. 2, Zhongshan N. Rd., Taipei 10449, Taiwan; leungwh22@gmail.com (W.-H.L.); b101091039@tmu.edu.tw (J.-S.C.); 2Ph.D. Program for Cancer Molecular Biology and Drug Discovery, College of Medical Science and Technology, Taipei Medical University and Academia Sinica, Taipei 11031, Taiwan; shihjw@tmu.edu.tw (J.-W.S.); d621108006@tmu.edu.tw (N.M.); 3Graduate Institute of Cancer Biology and Drug Discovery, College of Medical Science and Technology, Taipei Medical University, Taipei 11031, Taiwan; 4TMU Research Center of Cancer Translational Medicine, Taipei Medical University, Taipei 11031, Taiwan; 5Ph.D. Program for Translational Medicine, College of Medical Science and Technology, Taipei Medical University, Taipei 11031, Taiwan; 6Division of Colorectal Surgery, Department of Surgery, Taipei Medical University Hospital, Taipei Medical University, Taipei 110, Taiwan; poliwei@tmu.edu.tw; 7Department of Surgery, School of Medicine, College of Medicine, Taipei Medical University, Taipei 110, Taiwan; 8Division of General Surgery, Department of Surgery, Taipei Medical University Hospital, Taipei Medical University, Taipei 110, Taiwan

**Keywords:** colorectal cancer (CRC), cancer stem cell (CSC), metastasis, sulfasalazine, chemotherapy enhancement

## Abstract

Approximately 25% of colorectal cancer (CRC) patients will develop metastatic (m)CRC despite treatment interventions. In this setting, tumor cells are attracted to the epidermal growth factor receptor (*EGFR*) oncogene. Kirsten rat sarcoma (RAS) 2 viral oncogene homolog (*KRAS*) mutations were reported to drive CRC by promoting cancer progression in activating Wnt/β-catenin and RAS/extracellular signal-regulated kinase (ERK) pathways. In addition, *KRAS* is associated with almost 40% of patients who acquire resistance to EGFR inhibitors in mCRC. Multiple studies have demonstrated that cancer stem cells (CSCs) promote tumorigenesis, tumor growth, and resistance to therapy. One of the most common CSC prognostic markers widely reported in CRC is a cluster of differentiation 44 (CD44), which regulates matrix metalloproteinases 7/9 (MMP7/9) to promote tumor progression and metastasis; however, the molecular role of CD44 in CRC is still unclear. In invasive CRC, overexpression of MMP7 was reported in tumor cells compared to normal cells and plays a crucial function in CRC cetuximab and oxaliplatin resistance and distant metastasis. Here, we utilized a bioinformatics analysis and identified overexpression of *KRAS/MMP7/CD44* oncogenic signatures in CRC tumor tissues compared to normal tissues. In addition, a high incidence of mutations in *KRAS* and *CD44* were associated with some of the top tumorigenic oncogene’s overexpression, which ultimately promoted a poor response to chemotherapy and resistance to some FDA-approved drugs. Based on these findings, we explored a computational approach to drug repurposing of the drug, sulfasalazine, and our in silico molecular docking revealed unique interactions of sulfasalazine with the *KRAS/MMP7/CD44* oncogenes, resulting in high binding affinities compared to those of standard inhibitors. Our in vitro analysis demonstrated that sulfasalazine combined with cisplatin reduced cell viability, colony, and sphere formation in CRC cell lines. In addition, sulfasalazine alone and combined with cisplatin suppressed the expression of *KRAS/MMP7/CD44* in DLD-1 and HCT116 cell lines. Thus, sulfasalazine is worthy of further investigation as an adjuvant agent for improving chemotherapeutic responses in CRC patients.

## 1. Introduction

Colorectal cancer (CRC) remains a common cause of cancer-related fatalities globally [1], despite currently available advanced treatment options, including surgical resection, radiotherapy, chemotherapy, and targeted therapies [2]. CRC is often diagnosed in an advanced stage, with approximately 25% of patients displaying distant metastasis; this ultimately presents a challenge to clinicians [3,4,5]. Patients who undergo surgery and still exhibit an unresectable metastatic tumor are further treated with chemotherapy and targeted therapy [6]; however, these therapeutics only offer limited enhancement of overall survival for patients [7]. The molecular mechanisms of CRC are unclear, as the disease is heterogeneous. This poses a challenge in terms of patients’ responses to treatments [8]; therefore, there is an urgent need for novel therapeutic interventions aimed at combating metastatic (m)CRC [9]. Only 5% of mCRC patients reach the 5-year survival mark [10]. In this setting, tumor cells are attracted to the epidermal growth factor receptor (*EGFR*) oncogene, and then treatment options include cetuximab or panitumumab [11,12].

Several reports revealed that acquisition of Kirsten rat sarcoma 2 viral oncogene homolog (KRAS) mutations in CRC play significant roles in cancer progression [13,14]. KRAS is associated with almost 40% of patients who acquire resistance to EGFR inhibitors in mCRC [15,16]. Interestingly, others showed that amplification of the *KRAS* gene leads to activation of the Wnt/β-catenin and RAS/extracellular signal-regulated kinase (ERK) pathways [17,18]. Multiple studies demonstrated that tumor cells consist of subpopulations of cancer cells, known as cancer stem cells (CSCs), which promote tumorigenesis, tumor growth, and resistance to surgery, chemotherapy, and targeted therapies [19,20,21]. In addition, studies suggested that CSCs exhibit a population of functionally heterogeneous characteristics [22]. Distant metastasis into other organs was shown to be associated with these complex and diverse characteristics of metastatic colon stem cells [23]. One of the most common CSC prognostic markers widely reported in CRC is cluster of differentiation 44 (CD44), and multiple studies showed that CD44 activates mitogen-activated protein kinase (MAPK), phosphatidylinositol 3-kinase (PI3K)/Akt, and Wnt signaling pathways [24,25], and regulates activities of matrix metalloproteinases 7/9 (MMP7/9) to promote tumor progression and metastasis [26]; however, the molecular role of CD44 in CRC is still unclear [27,28].

MMP7 was highly expressed in invasive CRC compared to normal cells [29,30] and associated with distant metastasis [29]. MMP7 is expressed by epithelial tumor cells and has a key function in cancer progression [30,31]. In addition, others showed that MMP7 attaches to cell surface proteins and elevates metastasis progress [32]. In 2011, Ametller et al. demonstrated that MMP7 is upregulated after oxaliplatin resistance arises in CRC [33]. Additionally, MMP7 promotes metastasis and immune system invasion by tumor cells through interactions with the tumor microenvironment (TME), cancer cells, and the immune system [34,35]. These findings illustrate the urgent need for novel treatments for mCRC, which can be used either as monotherapy or with available therapeutics. In the present study, we explored a computational simulation to identify therapy-resistant oncogenes associated with distant metastasis in CRC. We also applied in silico molecular docking to predict interactions of sulfasalazine with CRC oncogenic signatures. Sulfasalazine is a niclosamide derivative anti-inflammatory drug that recently possesses anticancer properties against human tumors [36,37]. Here, we provide further mechanistic insights into sulfasalazine’s potential as an anti-CSC agent that can improve cisplatin treatment efficacy. 

## 2. Methods

### 2.1. Comparisons of KRAS/MMP7/CD44 Expressions in Normal, Tumor, and Metastatic Samples

To compare expressions of the *KRAS/MMP7/CD44* oncogenes between tumor, metastatic, and normal tissues, we explored the tumor, normal, and metastatic plot (TNMplot), (https://tnmplot.com/analysis/, accessed on 16 June 2021), an RNA-sequence-based rapid analysis which is used to compare data of selected genes [38]. Data were compared using the Kruskal–Wallis test, which is a method used to test samples originally from the same distribution of specimens followed by Dunn’s test, which assesses the significance of gene expressions in promoting CRC tumor metastasis, with *p* < 0.05 considered statistically significant.

### 2.2. Determining Associations of KRAS/MMP7/CD44 Mutations and Changes in Gene Expressions in CRC

Links between *KRAS/MMP7/CD44* mutations and changes in gene expressions in CRC were determined using the muTarget platform (https://www.mutarget.com/, accessed on 16 June 2021), a platform linking changes in gene expressions and the mutation status of solid tumors, based on a genotype analysis. These results can be used to identify alterations in genetic expressions and target analysis modules which can be used to identify genetic mutations [39]. Differences in expressions between the mutant group and wild-type group were considered statistically significant at *p* < 0.05.

### 2.3. Drug Sensitivity Analysis of the KRAS/MMP7/CD44 Oncogenes

To determine correlations between the *KRAS/MMP7/CD44* oncogenes and drug sensitivity of the genomics of drug sensitivity in cancer (GDSC) of the top 30 drugs in a pan-cancer database, we used the Gene Set Cancer Analysis (GSCA), a web-based tool used to analyze differentially expressed genes (DEGs) and correlations with drug sensitivity [40]. All drugs approved by the US Food and Drug Administration (FDA) were displayed in this analysis. To determine potential roles of the *KRAS/MMP7/CD44* oncogenes as factors that influence high diagnostic efficacy values in CRC patients, we explored an ROC analysis based on TCGA database. The response to chemotherapy treatment was based on RECIST criteria (*n* = 440) (Colorectal carcinoma (rocplot.org, accessed 17 June 2021)).

### 2.4. Survival Model Construction and Diagnostic Efficacy Evaluation of the KRAS/MMP7/CD44 Oncogenes

To assess the performance of the diagnostic efficacy using different cutoff points based on the sensitivity and specificity, we applied a receiver operating characteristic (ROC) curve. The curve was accordingly constructed by plotting the sensitivity (true positives) against false positives on the X- and Y-axes, respectively [41]. Furthermore, since diagnostic tests are not all equal, we evaluated whether the test measurement had specific conditions or not. We assessed the area under the ROC curve (AUC), and an AUC of 0.5 indicated no discrimination, while an AUC of 1.0 indicated discrimination of the curve that includes all possible decision thresholds from a diagnostic test result which were patients who had experienced disease onset and individuals who had not.

### 2.5. Protein-Protein Interaction (PPI) Analysis of KRAS/MMP7/CD44 Gene Signatures

In a further analysis, we applied the STRING tool (V11) (https://string-db.org/ accessed on, 18 June 2021), a web-based tool, which links proteins that cooperatively influence specific biological functions [42,43]. The STRING database was used under a high confidence of 0.772, and protein enrichment of *p* < 1.0 × 10^−16^ was obtained. Interactions among genes were analyzed according to correlations based on experimental data (pink), gene neighborhoods (green), gene fusion (red), gene co-occurrences (blue), and gene co-expressions (black). Moreover, a functional enrichment analysis and PPIs of the *KRAS/MMP7/CD44* oncogenes were constructed using the GSCA [40]. The gene ontology (GO) and Kyoto Encyclopedia of Genes and Genomes (KEGG) enrichment analyses were based on a gene set enrichment analysis (GSEA). The input gene sets were obtained from the STRING PPI network analysis, which included the 10 top genes.

### 2.6. Correlation Analysis of KRAS/MMP7/CD44 Expressions and Tumor Infiltration Levels

Correlations between *KRAS/MMP7/CD44* oncogenes expression and tumor infiltration levels were analyzed with an online bioinformatics tool, Tumor Immune Estimation Resource (TIMER) (https://cistrome.shinyapps.io/timer/ accessed on 7 October 2021), we determined correlations of *KRAS/MMP7/CD44* oncogenic signatures with a set of gene markers of immune infiltration cells including; B cells, CD8+ T cells, CD4+ cells and macrophages (with *p* < 0.05). The infiltration level was compared to the normal level using a two-sided Wilcoxon rank-sum test.

### 2.7. Binding Interactions of KRAS/MMP7/CD44 with Sulfasalazine

Ligand-receptor docking is a widely used computer simulation tool in drug discovery to predict binding energies of ligands [44]. We explored the CB-Dock tool, which predicts binding areas of a receptor, and calculates the binding distance and size [45]. The 3D structures of sulfasalazine (CID:5339), and standard inhibitors sotorasib (CID: 137278711), ilomastat/GM6001 (CID: 132519), and sorafenib (CID: 216239), respectively for *KRAS, MMP7*, and *CD44*, were retrieved from the PubChem database in spatial data file (SDF) format. Crystal structures of *KRAS* (PDB:6BP1), *MMP7* (PDB: 2Y6C), and *CD44* (PDB: 1UUH) were downloaded from the protein data bank as PDB files and used as docking models. The protein files (in PDB format) and ligand files (in SDF format) were uploaded and submitted to the CD-DOCK tool. Docking results were further visualized using discovery studio for analysis.

### 2.8. Cell Culture and Reagents

The DLD1 and HCT116 human colon cancer cell lines were acquired from American Type Culture Collection (ATCC, Manassas, VA, USA) and were cultured according to the vendor’s recommended conditions. All cells were maintained according to recommended culture conditions. Sulfasalazine was synthesized as described in a previous study [46], while Cisplatin (CDDP) was purchased from SelleckChem (Hsinchu, Taiwan). Stock solutions of sulfasalazine (10 mM) and cisplatin (10 mM) were prepared in dimethyl sulfoxide (DMSO; Sigma Aldrich, St. Louis, MO, USA) and stored at −20 °C.

### 2.9. CRC Colony Formation Assay

DLD-1 and HCT116 cells were seeded at 300 cells/well in 6-well plates (Corning) and treated with Sulfasalazine at 13.34 μM (DLD-1) and 12.38 μM (HCT116) and incubated for 7 days. The media was then removed, fixed and stained with a crystal violet solution (0.1% crystal violet, 1% methanol, and 1% formaldehyde) as described by Franken et al. [47]. Colonies were then quantified using a Cell3iMager neo scanner from the treated as compared to control colonies. 

### 2.10. Tumorsphere-Formation Assay

Tumorsphereformation assay was performed according to a previously described method [48] with some modifications. Accordingly, DLD-1 and HCT116 cell lines we seeded (3000 cells/well) in six-well ultra-low attachment plates (Corning, Corning, NY, USA) in serum-free media consisting of Dulbecco’s modified Eagle medium (DMEM)/Ham’s F12 (1:1), human epidermal growth factor (hEGF, 20 ng/mL). The cells were then allowed to aggregate and grow for at least a week. Cells (diameter > 50 μM), characterized by compact, non-adherent spheroid-like masses, were considered a tumorsphere, and counted with an inverted phase-contrast microscope.

### 2.11. ALDEFLUOR ALDH Activity Analysis

For the determination of aldehyde dehydrogenase (ALDH) activity analysis, we used the ALDEFLUORTM Kit (STEMCELL Technologies, Cambridge, UK) according to the manufacturer’s instructions. In short, 1 × 10^6^ colon cancer cells (DLD1 and HCT116) were first cultured with or without sulfasalazine (SSZ, 50 µM, 48 h) and followed by resuspension in 1 mL ALDEFLUOR buffer. The cells were washed in ALDEFLUOR buffer and maintained at 4 °C throughout the cell staining process. ALDH activity was determined using the fluorescence (FL1) and low side scatter (SCC) channels of a BD FACSCanto^TM^ flow cytometry system (BD Biosciences, CA, USA) and FACSDiva software (v 6.1.2, BD Biosciences, CA, USA). 

### 2.12. Sodium Dodecyl Sulfate-Polyacrylamide Gel Electrophoresis (SDS-PAGE) and Western Blot Analysis

Total protein lysates from CRC cells and tumor-spheres were extracted after different treatments. They were separated by SDS-PAGE using the Mini-Protean III system (Bio-Rad, Taiwan) and transferred onto polyvinylidene difluoride (PVDF) membranes using the Trans-Blot Turbo Transfer System (Bio-Rad). Membranes were incubated with the primary antibodies overnight at 4 °C. The following day, membranes were incubated with the secondary antibody. The proteins of interest were detected and visualized using enhanced chemiluminescence (ECL) detection kits (ECL Kits; Amersham Life Science, NJ, USA). Images were captured and analyzed using the UVP BioDoc-It system (Upland, CA, USA). 

### 2.13. Data Analysis

The Kruskal–Wallis test and Dunn’s test were used to assess expressions of *KRAS/MMP7/CD44* in tumor, normal, and metastatic tissues. The enrichment of GO and KEGG pathways was analyzed using gene counts and the false discovery rate (FDR). * *p* < 0.05 was accepted as being statistically significant.

## 3. Results

### 3.1. Identification of DEGs in CRC

Gene expression profiles (GEPs) from liver metastasis of CRC, and primary CRC samples as compared to the adjacent normal samples, obtained from different studies were extracted from the microarray dataset. Results were analyzed from three (3) different datasets, i.e., GSE14297, GSE49355, and GSE81558 datasets. The Venn diagram analysis showed 273, 30, and 51 overlapping upregulated genes from the above three databases, respectively (Figure 1A–C), Moreover, we used the volcano plots to further analyze the upregulated as compared to downregulated colon adenocarcinoma metastasis samples and normal samples, respectively (Figure 1D,F). The heatmap on (Figure 1G) displays the overexpressed overlapping genes from the selected datasets with *p*-value < 0.05 considered statistically significantly. 

### 3.2. Comparisons of KRAS/MMP7/CD44 Expressions in Normal, Tumor, and Metastatic Samples

We applied the TNM plot analysis to compare expressions of the *KRAS, MMP7*, and *CD44* genes in normal, tumor, and metastatic CRC samples from RNA-Seq data. Analytical results are presented as boxplots and violin plots. We compared the data by first using the Kruskal–Wallis test, which is a method used to test samples originally from the same distribution of specimens. The results acquired showed that increased expression levels of *KRAS*, *MMP7*, and *CD44* promoted primary tumor and metastasis in CRC samples compared to adjacent normal samples (Figure 2A–C). To compare if the samples were originally from the same distribution of specimens, we used the Kruskal–Wallis test, and results showed the same *p* value in both boxplots and violin plots as *p* = 3.88 × 10^−04^, *p* = 1.67 × 10^−148^, and *p* = 2.14 × 10^−123^ for *KRAS*, *MMP7*, and *CD44*, respectively (Figure 2D,F). We further used Dunn’s test which revealed the significance of *KRAS, MMP7*, and *CD44* expressions in promoting CRC tumor metastasis. More the Kaplan–Meier plot survival analysis showed that, high expression of *KRAS/MMP7/CD44* associated with poor overall survival in CRC, this validated that the samples originated from same distribution (Figure 2G–I). with *p* < 0.05 considered statistically significant. 

### 3.3. Linking KRAS/CD44 Mutations and Changes in Gene Expressions in CRC

Associations of *KRAS* mutations were first linked to changes in gene expressions in CRC at the genotypic level (GENOTYPE) using the muTarget tool. The top genes with higher expression levels linked to *KRAS* mutations compared to KRAS wild-type included *PHLDA1*, *HOXB6*, and *KCNN4*, which were associated with worse prognoses (Figure 3A). Moreover, we compared associations between changes in *KRAS* and *CD44* expressions to mutations of the top genes expressed in CRC at the target level (TARGET). The results we obtained showed that *ZFR*, *ABCB10*, and *OSMR* mutations were associated with higher *KRAS* expression levels, while *MAP3K10*, *TRIM71*, and *G3BP1* mutations were associated with higher CD44 expression levels in CRC compared to the wild-type. Patients with high expression levels of the *KRAS* and *CD44* oncogenes displayed poor prognoses compared to patients with low expression levels (Figure 3B,C). *p* < 0.05 was considered statistically significant.

### 3.4. Drug Sensitivity Analysis of the KRAS/MMP7/CD44 Oncogenes

We used the GSCA to determine the drug sensitivity of *KRAS, MMP*, and *CD44* as shown in circles (Figure 4). Correlation coefficients indicate that increased gene expression levels were resistant to the drug. Herein, high mRNA expression levels of *KRAS, MMP7*, and *CD44* (as indicated in orange circles) were associated with drug resistance. High expressions of the *KRAS, MMP7*, and *CD44* oncogenic signatures were potentially resistant to the drug. High expression of *KRAS* was demonstrated to be resistant to docetaxel, a chemotherapeutic drug [49], RDEA119, and selumetinib which are all MAPK kinase (MEK) inhibitors [50,51] (Figure 4A). In addition, increased expression levels of *MMP7* and *CD44* were shown to be resistant to Bx-912, which is a phosphoinositide-dependent kinase 1 (PDK1) inhibitor [52,53], navitoclax (a Bcl-2 inhibitor) [54,55], and vorinostat (a histone deacetylase (HDAC) inhibitor) [56,57] (Figure 4B,C).

### 3.5. Survival Model Construction and Diagnostic Efficacy Evaluation of the KRAS/MMP7/CD44 Oncogenes

To determine potential roles of the *KRAS/MMP7/CD44* oncogenes as factors that influence high diagnostic efficacy values in CRC patients, we explored an ROC analysis based on TCGA database. The response to chemotherapy treatment was based on RECIST criteria (*n* = 440). A boxplot of non-responder and responder values, as indicated by the box plot center line, evidently shows that expressions of the *KRAS/MMP7/CD44* genes in the colorectal adenocarcinoma (COAD) cohort from the ROC analysis revealed that these genes influenced the response of treatment by lowering the response rate of treatment in CRC patients. The ROC curve was based on true and false positive rates, and results showed that *KRAS, MMP7*, and *CD44* could serve as potential diagnostic biomarkers in CRC. The AUC scores of *KRAS, MMP7*, and *CD44* were 0.557, 0.537, and 0.551, respectively, with *p* < 0.05 considered significant (Figure 5A–F).

### 3.6. PPI Analysis of the KRAS/MMP7/CD44 Gene Signatures

The PPI network analysis among *KRAS/MMP7/CD44* gene signatures, using the STRING database under high confidence revealed interactions among the genes as indications of correlations based on experimental data (pink), gene neighborhoods (green), gene fusion (red), gene co-occurrences (blue), and gene co-expressions (black), e.g., KRAS with *CTNNB1, KRAS* with *CD44, CDH1* with *CD44, MMP7* with *CD44, MMP7* with *KRAS*, and *CTNNB1* with *MMP7.* The network had an initial four proteins, which were further increased to 24 nodes. Moreover, the network interactions showed that the genes were arranged in three different clusters: cluster 1 (red) included *KRAS, PIK3CA*, and *YES1*; cluster 2 (green) included *MMP7, CD44, CTNNB1*, and *EGFR;* and cluster 3 (blue) included *GSK3B, APC*, and *CTNNB1*. Protein enrichment of *p* < 1.0 × 10^−16^ was obtained from the clustering analysis, with a coefficient of 0.772. Table 1 is a summary of all the genes which interacted with the *KRAS, MMP7*, and *CD44* oncogenes with the confidence cutoff value set to 0.900 (Figure 6, Table 1).

Table 1 is a summary of all genes which interact with the *KRAS, MMP7*, and *CD44* oncogenes with a confidence cutoff value set to 0.900.

### 3.7. Functional Enrichment Analysis and PPI Construction of the KRAS/MMP7/CD44 Oncogenes

The GSCA was used to construct the GO and conduct the KEGG enrichment analysis, based on a GSEA. The input gene sets were obtained from the STRING PPI network analysis, which included 10 top genes (*APC, CD44, CTNNB1, EGFR, GSK3B, KRAS, MMP7, PIK3CA, SNAI1*, and *YES1*). These genes were largely involved in GO biological processes, including, movement of cells or subcellular components, locomotion, localization of cells, cell motility, and cell migration (Figure 7A). The KEGG pathways were mostly enriched in human papillomavirus infection, hepatocellular carcinoma, gastric cancer, breast cancer, CRC, and endometrial cancer (Figure 7B). All analyses were based on gene counts, and an FDR of <0.05 was considered statistically significant. To validate these finding, we used another network analytical tool called networkanalysist (https://www.networkanalyst.ca/ accessed, 20 June 2021), and identified similar pathways enriched from co-expressions of the *KRAS/MMP7/CD44* oncogenes (Figure 7C).

### 3.8. KRAS/MMP7/CD44 Oncogenes Expressions Were Correlated with Immune Cell Infiltration in CRC

To identify associations of *KRAS/MMP7/CD44* expressions with selected immune cells, we applied a correlation analysis between the above-mentioned oncogenes and immune infiltration cells (B cells, CD8+ T cells, CD4+ cells and macrophages), where markers were adjusted by purity. As anticipated, results showed correlations of immune cell markers in colorectal adenocarcinoma (COAD), specifically B cells, CD8+ T cells, CD4+ cells and M2 macrophages (Figure 8A–C), with *p* < 0.05 considered significant. Expressions of *KRAS/MMP7/CD44* were also found to be correlated with infiltrating levels of CD8+ T cells, CD8+ T cells, macrophages, and DCs (Figure 8D–F).

### 3.9. Molecular Docking Exhibited Putative Binding Energies for Sulfasalazine in Complex with MMP7, KRAS, and CD44

To determine how sulfasalazine binds to *the* MMP7, *KRAS, and CD44 proteins*, a molecular docking analysis was performed using the CD-Dock tool and discovery studio for analysis. Sulfasalazine showed high binding interactions with the crystal structures of KRAS [PDB: 6BP1], human MMP7 [PDB: 2Y6C], and CD44 [PDB: 1UUH], with respective binding affinities of −8.2, −7.9, and −7.2 kcal/mol (Figure 9A–C).The results were further visualized and interpreted using discovery studio. Results of the KRAS, MMP7, and CD44 proteins in complex with sulfasalazine revealed interactions by conventional hydrogen (H) bonds with ALA145 and ALA18 for KRAS, ALA186 for MMP7, and ARG78 for CD44. The interactions were further stabilized by van der Waals forces, π–π stacking interactions, and carbon-hydrogen bonds in the binding pockets of the receptors (Figure 9D,F). For further analysis, we compared interactions of sulfasalazine in complex with the above-mentioned oncogenes, with their standard inhibitors, i.e., sotorasib for *KRAS*, ilomastat/GM6001 *for MMP7*, and sorafenib for *CD44.* Interestingly the standard inhibitors in complex with KRAS and MMP7 displayed lower binding affinities of *−7.7 and −7.3 kcal/mol*, respectively, *and −7.9 kcal/mol* for CD44, which was slightly higher compared to those for sulfasalazine (Figure 10).

### 3.10. Sulfasalazine Suppressed the Tumorigenesis and Stemness of CRC Cells through the Downregulating KRAS, MMP7, and CD44 Signaling Axis

We investigated the inhibitory effects and therapeutic efficacy of Sulfasalazine (SSZ), against CRC cell lines. Using the SRB colorimetric assay, we showed SSZ treatment sensitized CRC cells towards cisplatin (CDDP) treatment (Figure 11A). Moreover, we tested the effects of SSZ on the biological properties of CRC cells and showed that SSZ treatment significantly reduced the clonogenicity, migration (Figure 11B) and tumorsphere-formation in both DLD-1 and HCT116 cells. ALDH is a cancer stemness marker, which is also shown to promote metastasis and treatment resistance [58], herein, we showed that compared to control or vehicle-treated cells, SSZ (50 μM, 48 h) suppressed ALDH activity in the DLD-1 cells by 11.64%, while in the HCT116 cells, it displayed a 5.5% reduction (Figure 11C). Mechanistically, western blot results supported our bioinformatics analysis as the expression of *KRAS, MMP7, and CD44* expression levels were reduced by SSZ alone and more prominently when combined with CDDP (Figure 11D).

## 4. Discussion

Despite recent developments of advanced therapies for CRC over the years, high morbidity and mortality continue to surge globally due to this disease [59]. CRC is often diagnosed at an advanced stage, and that ultimately limits efforts of chemotherapies and targeted therapies [7]. Approximately 25% of patients will develop mCRC [27], and that leads to poor clinical outcomes and challenges in treatment responses due to late detection of the disease [60]. Therefore, identifying and validating biomarkers in CRC patients will significantly contribute to early diagnoses and improved clinical outcome predictions [61]. There are no universal biomarkers for the identification of distant metastasis or resistance to current therapeutic drugs in CRC. *KRAS, BRAF*, and *PIK3CA* mutations have been widely reported in CRC progression, and their presence is associated with poor responses to anti-EGFR targeted therapies [62,63]. The EGFR plays a significant role in CRC cell progression and migration, and only 30% of patients respond to EGRF inhibitors, such as erlotinib, cetuximab, and panitumumab [64,65].

In the present study, we explored computational simulation to identify and validate biomarkers associated with CRC progression, distant metastasis, and resistance to chemotherapies and targeted therapeutics. We performed data mining from the NCBI (GEO) microarray datasets, and identified top upregulated oncogenes in CRC, and among others on the list, *KRAS, MMP7* and *CD44* we found to be upregulated in primary and metastatic CRC as compared to normal samples. Using the Kaplan–Meier plotter bioinformatics tool, we identified higher expression levels of the *KRAS, MMP7*, and *CD44* oncogenes in tumor and metastatic samples, compared to adjacent normal tissues. The RNA-Seq data were analyzed using the Kruskal–Wallis test and Dunn’s test which showed that samples were distributed from the same specimens, and the *KRAS, MMP7*, and *CD44* genes were more highly expressed in metastatic samples (Figure 1), and associated with poor overall survival. In addition, we found that a *KRAS* mutation was linked to higher expression levels of tumorigenic oncogenes of *PHLDA1*, *HOXB6*, and *KCNN4* among other top expressed genes compared to their expressions in *KRAS* wild-type at the genotypic level. Moreover, we compared the above analysis with the analysis at the target level, and results showed that increased levels of *KRAS* and *CD44* in CRC were associated with upregulation of *MAP3K10*, *TRIM71*, and *G3BP1* gene mutations, which promote proliferation, metastasis, and drug resistance [66,67,68]. Since expressions of the *KRAS/MMP7/CD44* oncogenes were demonstrated to promote distant metastasis and drug resistance in CRC [16,69], we further determined their responses to FDA-approved drugs when overexpressed in all cancers at the mRNA level, and results showed that when *KRAS* was elevated, it caused resistance to docetaxel, a chemotherapeutic drug [49], and to RDEA119 and selumetinib, which are both MEK inhibitors [50,51] (Figure 2A). 

In addition, increased expression levels of *MMP7* and *CD44* were shown to be related to resistance to Bx-912 (a PDK1 inhibitor) [52,53], navitoclax (a Bcl-2 inhibitor) [54,55], and vorinostat (an HDAC inhibitor) [56,57]. For further analysis, we determined the potential roles of the *KRAS/MMP7/CD44* oncogenes as factors that influence high diagnostic efficacy in CRC patients; therefore, we explored a ROC analysis based on the TCGA database, on patients’ responses to chemotherapy treatment based on RECIST criteria, and results showed that patients with increased levels of these genes responded poorly to treatment (Figure 3). To determine if the *KRAS/MMP7/CD44* oncogenes interacted with each other in cancer, we performed a PPI network analysis using STRING, and results showed co-expression and co-occurrences of these oncogenes within the same cluster; in addition, these genes were largely associated with enriched GO and KEGG pathways in CRC. The enriched GO biological processes included the movement of cells or subcellular components, locomotion, localization of cells, cell motility, and cell migration. At the same time, KEGG pathways were mostly enriched in human papillomavirus infection, hepatocellular carcinoma, gastric cancer, breast cancer, CRC, and endometrial cancer (Figure 5 and Figure 6).

Since increased expressions and high gene mutations of *KRAS/MMP7/CD44* were found to promote CRC progression, metastasis, and resistance to current chemotherapies and targeted therapies, we further explored a computational approach to drug repurposing of sulfasalazine, a niclosamide derivative anti-inflammatory drug, which was recently shown to possess anticancer properties against human tumors [36,37]. The drug was evaluated as a potential target for the *KRAS/MMP7/CD44* signaling pathway in mCRC. Computational medicine is an effective alternative that accelerates drug design and development while also reducing costs [70]. We applied molecular docking simulations to evaluate ligand-protein interactions. Interestingly, sulfasalazine exhibited high binding interactions with the crystal structure KRAS [PDB: 6BP1], MMP7 [PDB: 2Y6C], and CD44 [PDB: 1UUH], with respective Gibbs free binding energies of −8.2, −7.9, and −7.2 kcal/mol. These results were much higher than for FDA-approved drugs of sotorasib, ilomastat/GM6001, and sorafenib, which are standard inhibitors for KRAS, MMP7, and CD44 proteins, respectively, with binding affinities of −7.7, −7.3, and −7.9 kcal/mol, respectively, for CD44, which were slightly higher compared to sulfasalazine. These results revealed that sulfasalazine might be a potential KRAS/MMP7/CD44 signaling pathway inhibitor in mCRC. To further validate our bioinformatics prediction analysis, we performed in vitro analysis and demonstrated sulfasalazine alone and in combination with cisplatin successfully reduced cell viability, colony, and sphere formation, and reduced ALDH activities in CRC cell lines. In addition, the dose depended on treatment of sulfasalazine suppressed the expression of *KRAS/MMP7/CD44* in DLD-1 and HCT116 cell lines. Therefore, the efficacy of sulfasalazine is worthy of further investigation in vivo and in the preclinical setting as an alternative treatment in colon adenocarcinoma patients.

## 5. Conclusions

In summary, we identified *KRAS/MMP7/CD44* oncogenic signatures as major players in mCRC. Our bioinformatics analysis showed that these oncogenes are overexpressed in CRC tumor tissues compared to normal tissues, resulting in tumor progression, metastasis, poor prognoses, and resistance to the current chemotherapies and targeted therapies. Additionally, high mutations in the *KRAS* and *CD44* genes were associated with over-expressions of some of the top tumorigenic oncogenes, which ultimately promoted poor responses to chemotherapy and resistance to some FDA-approved drugs. Our in vitro analysis demonstrated that sulfasalazine alone and in combination with cisplatin successfully reduced cell viability, colony, and sphere formation in CRC cell lines. In addition, the dose depended on treatment of sulfasalazine suppressed the expression of *KRAS/MMP7/CD44* in DLD-1 and HCT116 cell lines. Therefore, the efficacy of sulfasalazine is worthy of further investigation in vivo and in the preclinical setting as an alternative treatment in colon adenocarcinoma patients.

## Figures and Tables

**Figure 1 biomedicines-10-00377-f001:**
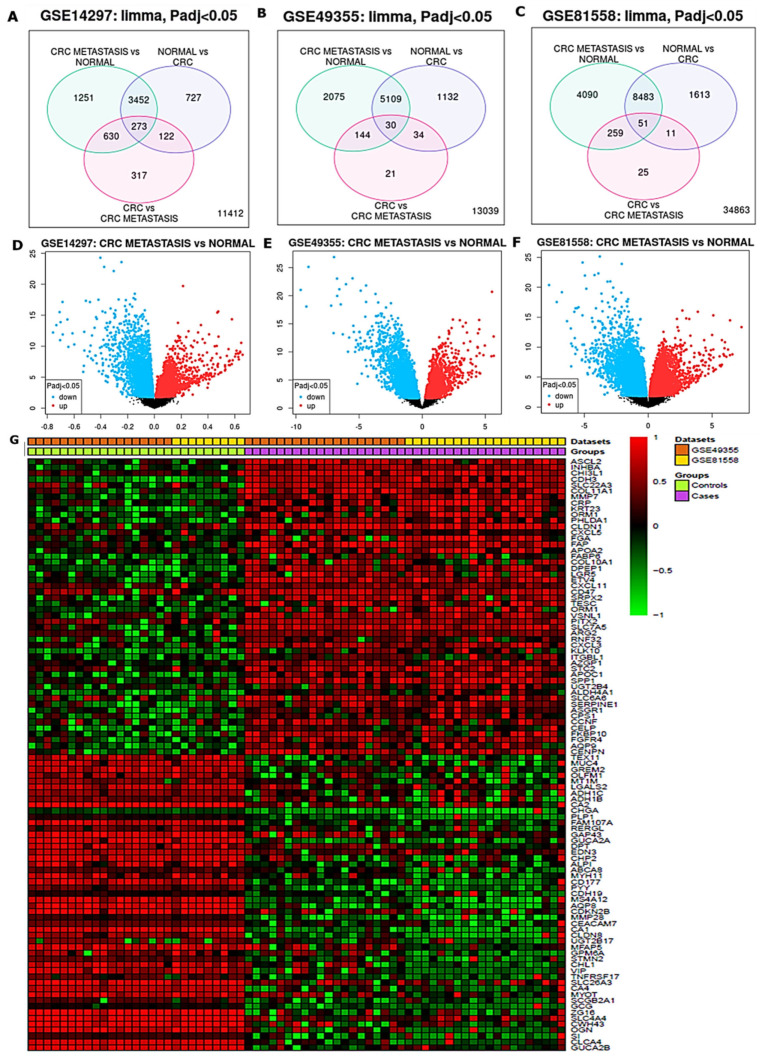
Differentially expressed genes (DEGs) in liver metastatic colon adenocarcinoma (CRC), primary CRC and normal sample, all obtained from GSE14297, GSE49355 and GSE81558 datasets. (**A**–**C**) Venn diagram showed 273, 30 and 51 overlapping upregulated genes from the above three databases, respectively. (**D**–**F**) volcano plots showing upregulated (red) and downregulated (blue) from metastatic samples as compared normal samples, respectively (**G**) shows the heatmap of overexpressed overlapping genes. with *p*-value < 0.05 considered statistically significantly.

**Figure 2 biomedicines-10-00377-f002:**
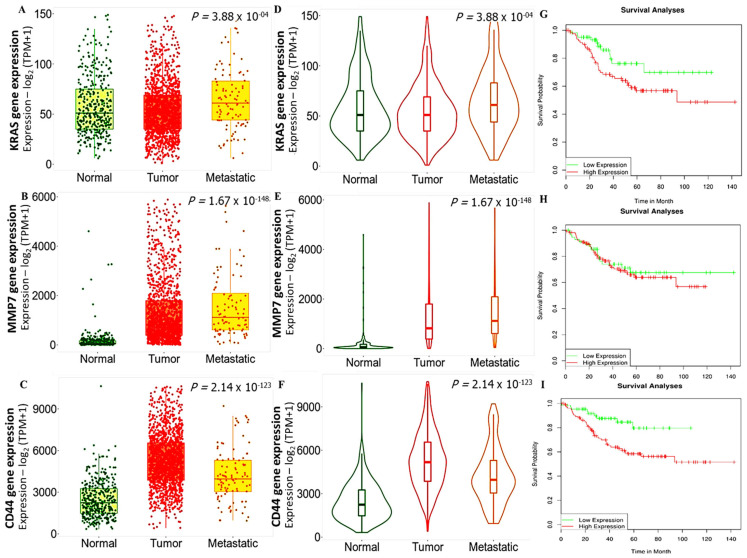
Increased expressions of the *KRAS, MMP7*, and *CD44* oncogenes were associated with colorectal cancer (CRC) tumor and distant metastasis. (**A**–**C**) Boxplots and (**D**–**F**) violin plots of increased levels of the *KRAS, MMP7*, and *CD44* oncogenes in CRC tumor and metastatic samples compared to normal samples. The analysis was based on the Kruskal–Wallis test and Dunn’s test. (**G**–**I**) Kaplan–Meier plot survival analysis showing that, the overexpression of *KRAS/MMP7/CD44* associated with poor overall survival in CRC, with *p* < 0.05 considered statistically significant.

**Figure 3 biomedicines-10-00377-f003:**
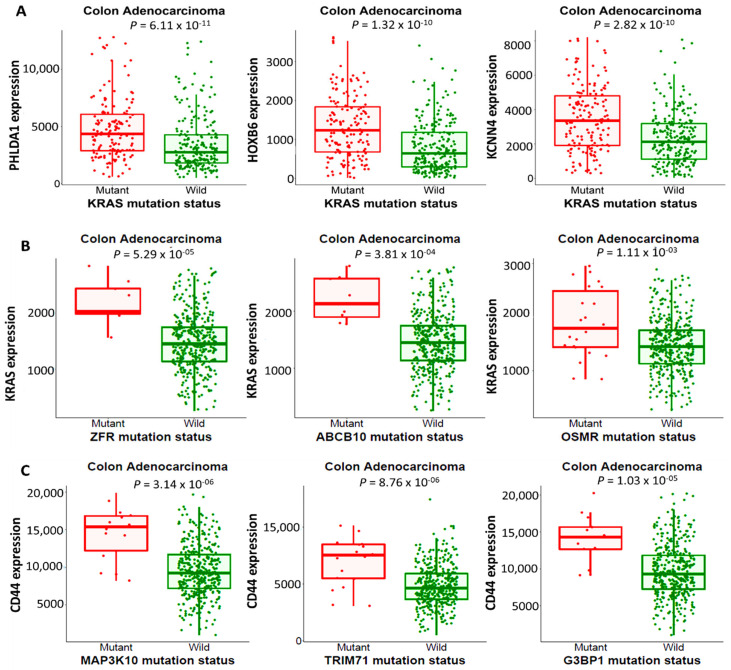
*KRAS* and *CD44* mutations were associated with worsened clinical outcomes in colorectal cancer (CRC). (**A**) Associations of *KRAS* mutations with the top three highly expressed genes (*PHLDA1*, *HOXB6*, and *KCNN4*) in CRC. (**B**,**C**) The top three genes (*ZFR*, *ABCB10*, and *OSMR*) for *KRAS* and (*MAP3K10*, *TRIM71*, and *G3BP1*) for *CD44* had stronger mutations and were associated with changes in expressions of *KRAS* and *CD44* in CRC, with *p* < 0.05 considered significan.

**Figure 4 biomedicines-10-00377-f004:**
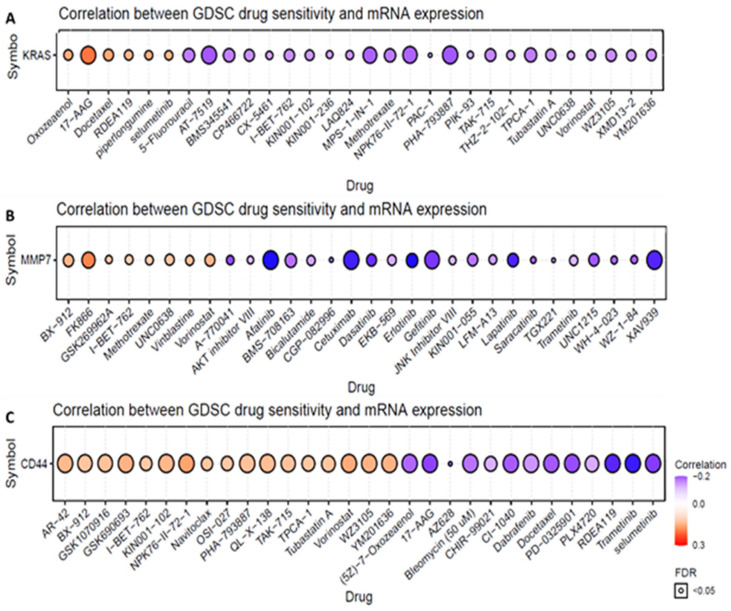
Drug sensitivity of the *KRAS/MMP7/CD44* oncogenes from GSCA. (**A**–**C**) Correlations between genomics of drug sensitivity in cancer (GDSC) from FDA-approved drugs. Positive Spearman correlation coefficients (orange circles) indicate that increased gene expression levels were resistant to the drug, compared to negative correlations shown in blue, which indicate sensitivity to the drug.

**Figure 5 biomedicines-10-00377-f005:**
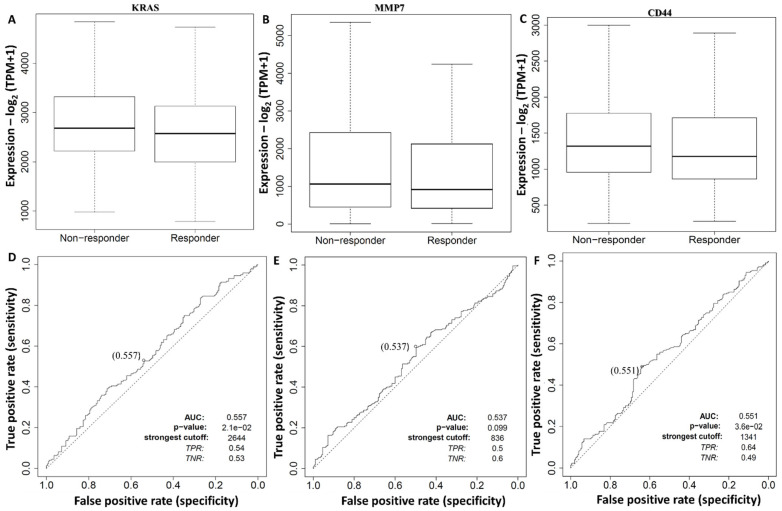
Diagnostic receiver operating characteristic (ROC) curves of the *KRAS/MMP7/CD44* oncogenes distinguishing colorectal adenocarcinoma (COAD) responding and non-responding patients. (**A**–**C**) Boxplot center lines showing expressions of the *KRAS/MMP7/CD44* genes in a COAD cohort from an ROC analysis revealing the influence on the responses of these genes to treatment by lowering the response rate. (**D**–**F**) ROC curves based on true and false positive rates. The area under the curve (AUC) scores of *KRAS, MMP7*, and *CD44* were 0.557, 0.537, and 0.551, respectively, with *p* < 0.05 considered significant. This suggests that *KRAS, MMP7*, and *CD44* can potentially serve as diagnostic biomarkers in colorectal cancer.

**Figure 6 biomedicines-10-00377-f006:**
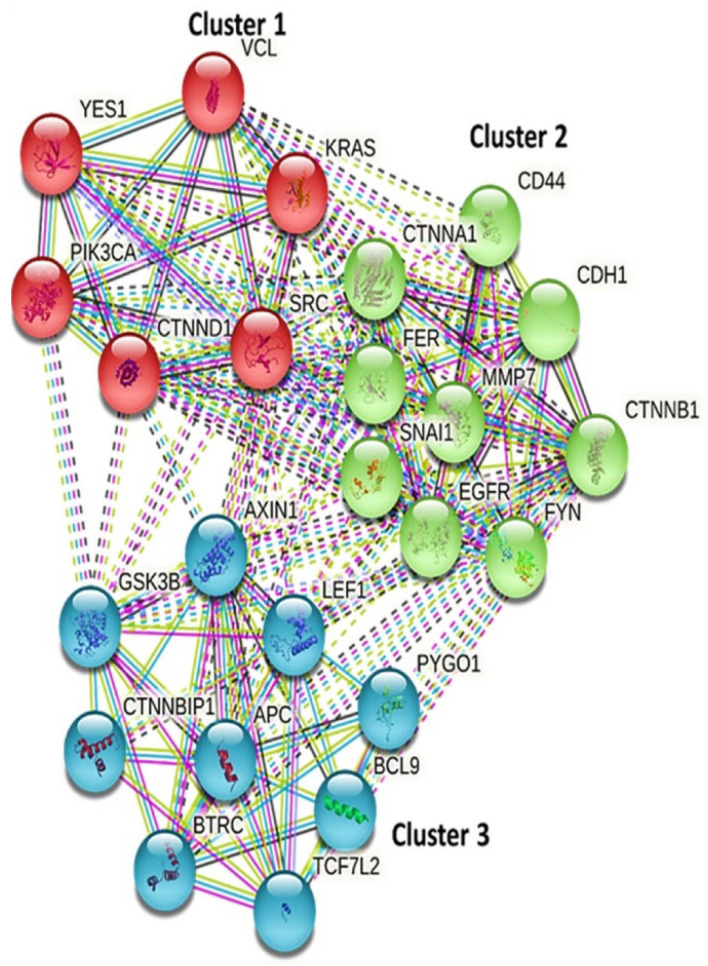
STRING database predicted protein-protein interacting networks (PINs) among oncogenic markers of the *KRAS/MMP7/CD44* gene signatures. Three clusters of interacting networks were constructed from the three genes. Furthermore, interactions among the correlated genes are displayed based on experimental data (pink), gene neighborhoods (green), gene fusion (red), gene co-occurrences (blue), and gene co-expressions (black). Associations were detected of KRAS with *CTNNB1, KRAS* with *CD44, CDH1* with *CD44, MMP7* with *CD44, MMP7* with *KRAS*, and *CTNNB1* with *MMP7*. Moreover, protein enrichment of *p* < 1.0 × 10^−16^ was obtained from the clustering analysis, with a coefficient of 0.772.

**Figure 7 biomedicines-10-00377-f007:**
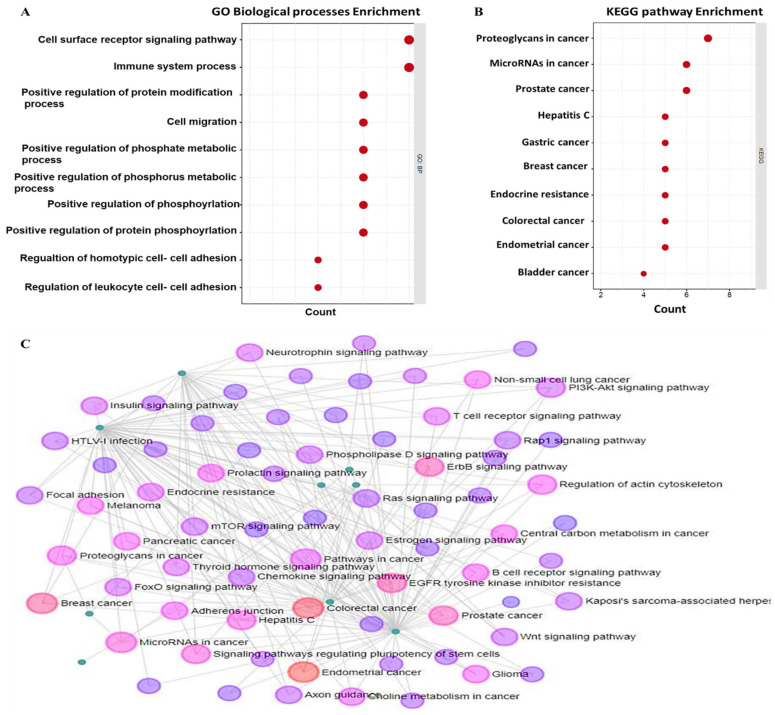
Gene ontology (GO) and Kyoto Encyclopedia of Genes and Genomes (KEGG) pathway enrichment of the *KRAS/MMP7/CD44* oncogenes. (**A**) Biological processes of the top 10 terms; results were based on gene counts and a false discovery rate (FDR) of <0.05. The most involved terms included movement of cells or subcellular components, locomotion, localization of cells, cell motility, and cell migration. (**B**) KEGG pathway of the top 10 enriched pathways. This analysis was also based on gene counts and an FDR of <0.05. (**C**) Further KEGG pathway enrichment analysis from a different tool showing that co-expressed genes exhibited enrichment in colorectal cancer, breast cancer, and endometrial cancer (red).

**Figure 8 biomedicines-10-00377-f008:**
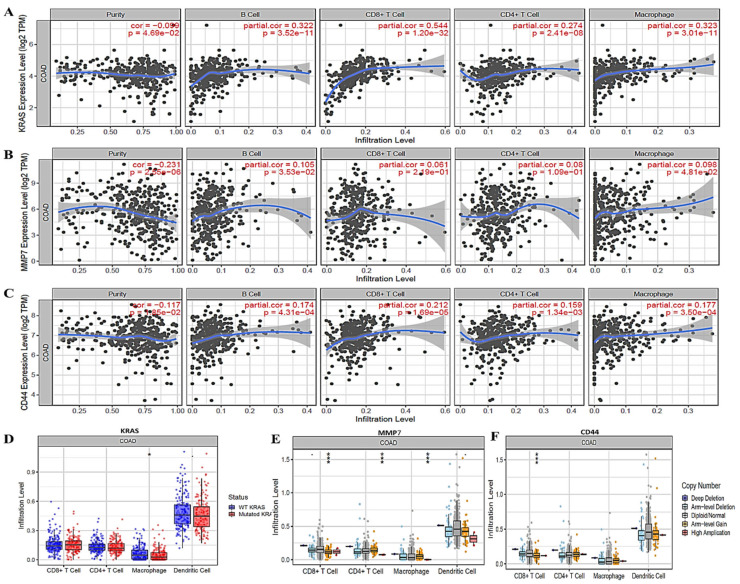
*KRAS/MMP7/CD44* expressions correlated with immune infiltrating cells in (COAD). (**A**) *KRAS*, (**B**) *MMP7*, and (**C**) *CD44* expression levels displayed associations with tumor purity and were positively correlated with infiltrating levels of B cells, CD8+ T cells, CD4+ cells and M2 macrophages, (**D**) *KRAS*, (**E**) *MMP7*, and (**F**) *CD44* expressions were correlated with abundances of tumor infiltrates including CD8+ T cells, CD4+ T cells, macrophages, and dendritic cells in CRC. Red represents considerable positive correlations with high amplification, while blue represents negative correlations. The infiltration level was compared to the normal level using a two-sided Wilcoxon rank-sum test. *p*-value Significant Codes: 0 ≤ *******
*p* < 0.001 ≤ ****** *p* < 0.01 ≤ *****
*p* < 0.05 ≤ *p* < 0.1.

**Figure 9 biomedicines-10-00377-f009:**
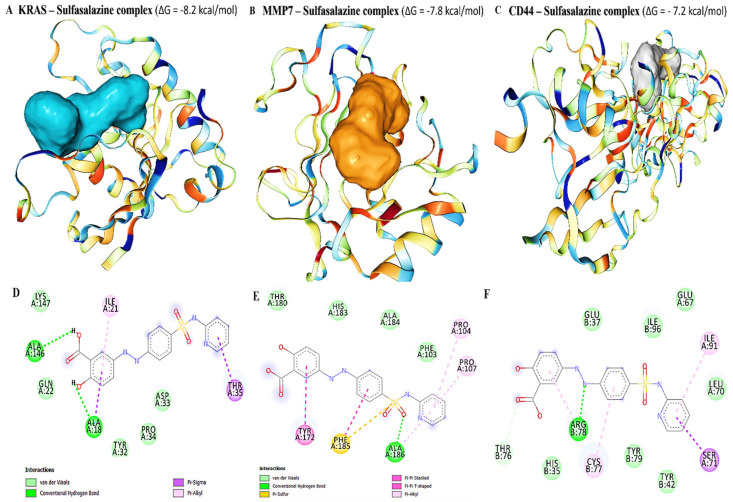
Sulfasalazine in complex with *KRAS, MMP7*, and *CD44* displayed putative binding interactions. (**A**–**C**) Binding interactions of *KRAS*, *MMP7*, and *CD44*, with sulfasalazine exhibited binding affinities of −8.2, −7.9, and −7.2 kcal/mol, respectively. Moreover, the visualization analysis revealed interactions by conventional hydrogen bonds (H-bonds) with ALA145 and ALA18 for *KRAS*, ALA186 for MMP7, and ARG78 for *CD44*. Interactions were further stabilized by van der Waals forces, π–π stacking interactions, and carbon-hydrogen bonds in the binding pockets of the receptors (**D**–**F**).

**Figure 10 biomedicines-10-00377-f010:**
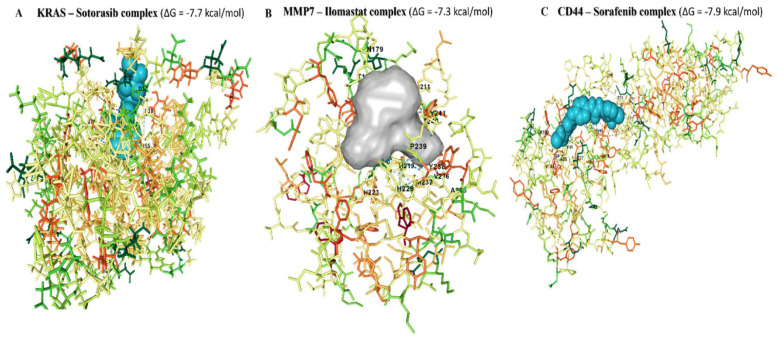
Binding interactions of *KRAS/MMP7/CD44* with standard inhibitors. (**A**) Sotorasib bound to *KRAS* with a binding energy of −7.7 kcal/mol. (**B**) Ilomastat interacted with *MMP7* with a binding energy of −7.3 kcal/mol, while CD44 in complex with sorafenib exhibited a binding energy of −7.9 kcal/mol (**C**). These results revealed the potential anticancer activities of sulfasalazine as a target drug for *KRAS/MMP7/CD44* signaling pathways in colorectal cancer.

**Figure 11 biomedicines-10-00377-f011:**
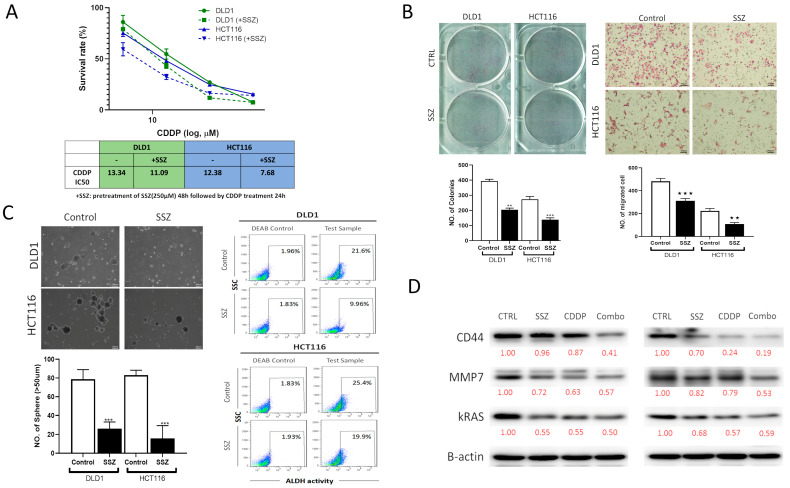
Sulfasalazine (SSZ) treatment reduced the tumorigenic properties of CRC cells and enhanced cisplatin (CDDP) efficacy. (**A**) SSZ enhanced cisplatin efficacy in both the DLD-1 and HCT116 cell lines. IC50 values are shown. Representative micrographs for the suppressive effects of SSZ on the ability of DLD-1 and HCT116 cells to form (**B**) colonies and migration, and (**C**) tumorspheres, as well as showing reduced ALDH activity (cancer stemness marker) in DLD-1 and HCT116 cells. (**D**) Western blot results show that SSZ and CDDP combined treatment significantly reduced the expression level of KRAS, MMP7, and CD44 on CRC cells compared to their vehicle-treated counterparts. β-actin served as the loading control. ** *p* < 0.01, *** *p* < 0.001. Numbers in red represent the relative expression level of the band intensity estimated using ImageJ software.

**Table 1 biomedicines-10-00377-t001:** Genes interacting with *KRAS, MMP7* and *CD44* oncogenes.

**KRAS**		
**Interactive Genes**	**Accession Number**	**Score**
APC	ENSP00000257430	0.652
CD44	ENSP00000398632	0.504
CTNNB1	ENSP00000344456	0.756
EGFR	ENSP00000256078	0.998
GSK3B	ENSP00000256078	0.674
MMP7	ENSP00000260227	0.427
**MMP7**		
**Interactive genes**	**Accession number**	**Score**
CD44	ENSP00000398632	0.972
CTNNB1	ENSP00000344456	0.966
EGFR	ENSP00000275493	0.71
KRAS	ENSP00000256078	0.427
SNAI1	ENSP00000244050	0.545
AXIN1	ENSP00000262320	0.561
**CD44**		
**Interactive genes**	**Accession number**	**Score**
CTNNB1	ENSP00000344456	0.664
EGFR	ENSP00000275493	0.921
KRAS	ENSP00000256078	0.568
MMP7	ENSP00000260227	0.972
LEF1	ENSP00000265165	0.627
FYN	ENSP00000346671	0.778

## Data Availability

The datasets generated and/or analyzed in this study are available on reasonable request.

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
