# Peer review of "Preclinical Identification of Sulfasalazine’s Therapeutic Potential for Suppressing Colorectal Cancer Stemness and Metastasis through Targeting KRAS/MMP7/CD44 Signaling"

_biomedicines, 2022, doi:10.3390/biomedicines10020377_

Round 1

Reviewer 1 Report

My opinion:

  • The very interesting results were obtained after bioinformatics analysis,
  • The conclusion regarding futher in vivo studies of sulfasalazine is important,

Author Response

Journal Name: BIOMEDICINES

Manuscript Title: Preclinical identification of Sulfasalazine’s therapeutic potential for suppressing colorectal cancer stemness and metastasis through targeting KRAS/MMP7/CD44 signaling.

Dear Editor,

Thank you for your comments and suggestions on my manuscript to this highly reputable journal. The manuscript has been extensively revised based on your advice and the comments and suggestions of the referees. We have modified the manuscript accordingly, and detailed corrections are listed below point by point:

Finally, as advised we have outlined above, a point-by-point response to each of the referee’s comments with cogent scientific explanations. We hope that the revised manuscript, after the incorporation of these changes, would now be formally accepted in the BIOMEDICINES journal. We look forward to your positive response.

Yours sincerely

Mr Ntlotlang Mokgautsi

Reviewer 2 Report

In this manuscript, the authors have identified overexpression of KRAS/MMP/CD44 in colorectal cancer cells using bioinformatic analyzes. Using computational analyzes, the authors found that sulfasalazine has high binding affinities with KRAS/MMP/CD44 than other standard inhibitors. The sulfasalazine inhibit cancer cell survival and suppresses KRAS/MMP/CD44. This is an interesting study. But, the conclusions are not well supported by the data. The authors should address the below points.

  1. The novelty of the study should be clearly emphasized.
  2. The abstract should be reduced.
  3. Language editing is needed. The manuscript should be checked meticulously for spelling mistakes.
  4. The units for gene expression on Y-axis in Figure 2 A-F is needed.
  5. There are two Figure 2 in the manuscript. 
  6. The units for gene expression on Y-axis in Figure 4 is needed.
  7. In Figure 10, the Western blot images of MMP7 and CD44 are not convincing and must be replaced with good images. Densitometry analysis of Western blotting data is needed.
  8. The authors should include in vitro data to support their claim that sulfasalazine suppresses metastasis of CRC cells. The in vivo data will be ideal.
  9. More appropriate data are required to support the role of sulfasalazine in suppressing colorectal cancer stemness. Immunofluorescence can be useful.

Author Response

Journal Name: BIOMEDICINES

Manuscript Title: Preclinical identification of Sulfasalazine’s therapeutic potential for suppressing colorectal cancer stemness and metastasis through targeting KRAS/MMP7/CD44 signaling.

Dear Editor,

Thank you for your comments and suggestions on my manuscript to this highly reputable journal. The manuscript has been extensively revised based on your advice and the comments and suggestions of the referees. We have modified the manuscript accordingly, and detailed corrections are listed below point by point:

In this manuscript, the authors have identified overexpression of KRAS/MMP/CD44 in colorectal cancer cells using bioinformatics analyzes. Using computational analyzes, the authors found that sulfasalazine has high binding affinities with KRAS/MMP/CD44 than other standard inhibitors. The sulfasalazine inhibits cancer cell survival and suppresses KRAS/MMP/CD44. This is an interesting study. But, the conclusions are not well supported by the data. The authors should address the below points.

Comments and responses:

  1. The novelty of the study should be clearly emphasized.
  • Novelty of the study emphasized.
  1. The abstract should be reduced.
  • Abstract modified and reduced.
  1. Language editing is needed. The manuscript should be checked meticulously for spelling mistakes.
  • Language edited, and possible errors checked, and corrected.
  1. The units for gene expression on Y-axis in Figure 2 A-F is needed.
  • Units added.
  1. There are two Figure 2 in the manuscript. 
  • Labeling error corrected, thanks.
  1. The units for gene expression on Y-axis in Figure 4 is needed.
  • Figure modified
  1. In Figure 10, the Western blot images of MMP7 and CD44 are not convincing and must be replaced with good images. Densitometry analysis of Western blotting data is needed.
  • Western blots were replaced with better resolution images, and quantification was done on the blots, thanks.
  1. The authors should include in vitro data to support their claim that sulfasalazine suppresses metastasis of CRC cells. The in vivo data will be ideal.
  • In vitro studies were included, thanks
  1. More appropriate data are required to support the role of sulfasalazine in suppressing colorectal cancer stemness. Immunofluorescence can be useful.
  • Thanks for the comment we will add these experiment in our future paper with sulfasalazine as a target drug.

  1. Finally, as advised we have outlined above, a point-by-point response to each of the referee’s comments with cogent scientific explanations. We hope that the revised manuscript, after the incorporation of these changes, would now be formally accepted in the BIOMEDICINES We look forward to your positive response.

Yours sincerely

Mr Ntlotlang Mokgautsi

Round 2

Reviewer 2 Report

The authors have mentioned that all the comments were addressed in the response letter. But, the following concerns are not addressed to support their conclusions.

  1. The units for expression analyzes on Y-axis in Figure 4 is needed.
  2. In Figure 10, the authors didn't provide a good image for MMP7 and CD44. Even, the CD44 Western blot image was not changed.  
  3. The authors didn't provide the appropriate in vitro data to support their claim that sulfasalazine suppresses metastasis of CRC cells. There is no explanation about this point as well. 
  4. More appropriate data to support the role of sulfasalazine in suppressing colorectal cancer stemness was not provided. There is no explanation for this as well.
